# Fridamycin A, a Microbial Natural Product, Stimulates Glucose Uptake without Inducing Adipogenesis

**DOI:** 10.3390/nu11040765

**Published:** 2019-04-01

**Authors:** Sun-Young Yoon, Seoung Rak Lee, Ji Young Hwang, René Benndorf, Christine Beemelmanns, Sang J. Chung, Ki Hyun Kim

**Affiliations:** 1School of Pharmacy, Sungkyunkwan University, Suwon 16419, Korea; youcan26@skku.edu (S.-Y.Y.); davidseoungrak@gmail.com (S.R.L.); smailemaster@naver.com (J.Y.H.); 2Leibniz Institute for Natural Product Research and Infection Biology—Hans-Knöll-Institute, Beutenbergstraße 11a, 07745 Jena, Germany; rene.benndorf@hki-jena.de (R.B.); Christine.beemelmanns@hki-jena.de (C.B.)

**Keywords:** *Actinomadura* sp. RB99, fridamycin A, 3T3-L1 cells, type 2 diabetes, glucose uptake

## Abstract

Type 2 diabetes is a complex, heterogeneous, and polygenic disease. Currently, available drugs for treating type 2 diabetes predominantly include sulfonylureas, α-glucosidase inhibitors, and biguanides. However, long-term treatment with these therapeutic drugs is often accompanied by undesirable side effects, which have driven interest in the development of more effective and safer antidiabetic agents. To address the urgent need for new chemical solutions, we focused on the analysis of structurally novel and/or biologically new metabolites produced by insect-associated microbes as they have recently been recognized as a rich source of natural products. Comparative LC/MS-based analysis of *Actinomadura* sp. RB99, isolated from a fungus-growing termite, led to the identification of the type II polyketide synthase-derived fridamycin A. The structure of fridamycin A was confirmed by ^1^H NMR data and LC/MS analysis. The natural microbial product, fridamycin A, was examined for its antidiabetic properties in 3T3-L1 adipocytes, which demonstrated that fridamycin A induced glucose uptake in 3T3-L1 cells by activating the AMP-activated protein kinase (AMPK) signaling pathway but did not affect adipocyte differentiation, suggesting that the glucose uptake took place through activation of the AMPK signaling pathway without inducing adipogenesis. Our results suggest that fridamycin A has potential to induce fewer side effects such as weight gain compared to rosiglitazone, a commonly used antidiabetic drug, and that fridamycin A could be a novel potential therapeutic candidate for the management of type 2 diabetes.

## 1. Introduction

Type 2 diabetes is a metabolic disease characterized by abnormally high blood glucose levels and cellular insulin resistance despite normal insulin production by the pancreas [1]. Persistent hyperglycemia is associated with cardiovascular disorders, renal dysfunction, and retinopathy [2]. In an insulin-resistant state, insulin cannot activate the insulin signaling pathway to stimulate glucose uptake in insulin-sensitive tissues such as adipose tissues, liver, and skeletal muscle [3,4]. Thiazolidinediones, such as rosiglitazone, metformin, and glyburide, have been used as insulin-sensitizing drugs for the treatment of type 2 diabetes [5,6]. As long-term treatment with these hypoglycemic agents is often associated with adverse effects including peripheral vascular disease, gastrointestinal events, cardiovascular diseases, weight gain, and edema, the development of new antidiabetic agents from natural sources which have fewer side effects than commonly used drugs has become necessary. Recently, natural products have been reported to improve insulin sensitivity via the activation of AMP-activated protein kinase (AMPK), which is believed to be a therapeutic target for the treatment of type 2 diabetes [4,7,8].

Insect-associated microbes have recently been recognized as an untapped natural source of structurally and biologically novel metabolites [9,10]. Our group has performed extensive chemical investigations of bioactive secondary metabolites derived from insect-associated microbes and identified cytotoxic beauvetetraones A−C (phomaligadione-derived polyketide dimers) from the entomopathogenic fungus *Beauveria bassiana* [11], neuroprotective isoflavonoids from a termite-associated *Streptomyces* sp. RB1 [12], and antibacterial macrotermycins A−D (20-membered, glycosylated, polyketide macrolactams) from a termite-associated actinomycete, *Amycolatopsis* sp. M39 [13]. As part of our continuing objective to discover biologically therapeutic natural products [14,15,16,17,18], we focused on the termite-associated *Actinomadura* sp. RB99, isolated from the fungus-growing termite *Macrotermes natalensis*. Our LC/MS/UV-based dereplication strategy identified novel bioactive natural products, natalenamide A−C (compounds 1–3), which have skin-whitening effects [19]. Further chemical investigation of MeOH extracts of *Actinomadura* sp. RB99 using a comparative LC/MS-based analytical approach led to the purification of the type II polyketide synthase-derived fridamycin A.

Here, we report the isolation and chemical identification of fridamycin A and the studies performed to examine its antidiabetic properties in 3T3-L1 adipocytes. Our cell-based studies indicated that fridamycin A could be a potential new therapeutic candidate for the treatment of type 2 diabetes. 

## 2. Materials and Methods

### 2.1. General Experimental Procedures

Optical rotations were calculated using a Jasco P-1020 polarimeter (Jasco, Easton, MD, USA). UV spectra were acquired on an Agilent 8453 UV-visible spectrophotometer (Agilent Technologies, Santa Clara, CA, USA). NMR spectra were acquired using a Varian UNITY INOVA 800 NMR spectrometer (Varian, Palo Alto, CA, USA) operating at 800 MHz (^1^H) with chemical shifts reported in ppm (δ). Preparative high-performance liquid chromatography (HPLC) was performed using a Waters 1525 Binary HPLC pump with a Waters 996 Photodiode Array Detector (Waters Corporation, Milford, CT, USA) and the column temperature was maintained at 30 °C. The mobile phase consisted of H_2_O (A) and CH_3_CN (B) with a gradient system as follows: 10–100% B (0–60 min); 100% B (60–71 min); 100–10% B (71–72 min), followed by 15 min of reconditioning. Silica gel 60 (Merck, 230–400 mesh) and RP-C18 silica gel (Merck, 230–400 mesh) were used for column chromatography. Semi-preparative HPLC was performed using a Shimadzu Prominence HPLC System with an SPD-20A/20AV Series Prominence HPLC UV-Vis Detector (Shimadzu, Tokyo, Japan) and the column temperature was maintained at 30 °C. The mobile phase consisted of H_2_O (A) and MeOH (B) with an isocratic system as follows: 67% B (0–60 min); 67–100% B (60–61 min); 100% B (61–71 min), followed by 12 min of reconditioning. LC/MS analyses were performed using an Agilent 1200 Series HPLC system (Agilent Technologies, Santa Clara, CA, USA) equipped with a diode array detector and a 6130 Series electrospray ionization (ESI) mass spectrometer with an analytical Kinetex (4.6 × 100 mm, 3.5 μm) HPLC column. The mobile phase consisted of H_2_O (A) and MeOH (B) with a gradient system as follows: 10–100% B (0–30 min); 100% B (30–40 min); 100–10% B (40–41 min); 10% B (41–50 min). The flow rate of the mobile phase was 0.3 mL/min and the column temperature was maintained at 40 °C. A 5 µL aliquot from a 100 µL of sample (0.01 mg/µL) was used in the negative-mode ESI-MS at *m/z* 100–1000 Da range with acquisition times of 0.2 s in the centroid mode. The ESI conditions were set as follows: capillary voltage 2.0 kV, convoltage 50 V, source temperature 120 °C, desolvation temperature 350 °C, and desolvation gas flow 800 L/h. High purity nitrogen gas was used as the nebulizer and auxiliary gas. The collision energy for the detection of the precursor ions was set to 3 eV. Merck pre-coated silica gel F254 plates and RP-18 F254s plates were used for thin-layer chromatography (TLC). Spots were detected on TLC plates under UV light or by heating after spraying with anisaldehyde-sulfuric acid.

### 2.2. Microbial Material

*Actinomadura* sp. RB99 was isolated from the surface of a termite worker of the species *M. natalensis* (colony Mn105, S24 40 30.5 E28 47 50.4) in January 2010. 

### 2.3. DNA Extraction and PCR Amplification

*Actinomadura* sp. RB99 was grown in nutrient-rich liquid ISP2 media, cells were harvested, and genomic DNA was extracted using a GenJet genomic DNA purification kit (Thermo Scientific, Waltham, MA, USA, #K0721) following the manufacturer’s instructions. For phylogenetic studies, the 16S rRNA gene was amplified using the primer set 1492R/27F.

### 2.4. Sequencing and Species Identification

Sequences were assessed for purity and mismatches using BioEdit. Resulting sequences were deposited in GenBank (accession number: KY558684). Blast analyses with nearly-complete 16S rRNA sequences (1368 bp) were performed using the NCBI database (reference RNA sequences). Two different phylogenetic trees were reconstructed with neighbor-joining or maximum-likelihood algorithms using MEGA software version 7.0.26. 

### 2.5. Extraction and Isolation

*Actinomadura* sp. RB99 was grown in 50 mL of ISP-2 broth for 7 days at 30 °C (pre-culture) and used to inoculate 100 ISP-2 agar plates. Plates were incubated for 10 days at 30 °C, cut into small pieces, consolidated, and immersed overnight in MeOH. The MeOH phase was filtered and evaporated under reduced pressure. The MeOH extract (20 g) was dissolved in distilled water (700 mL) and then solvent partitioned with EtOAc (700 mL) three times, resulting in 1.1 g of residue. The EtOAc-soluble fraction (1.1 g) derived from the MeOH extract was loaded onto a silica gel (230–400 mesh) column for chromatographic separation and eluted with a gradient solvent system of CH_2_Cl_2_–MeOH (100:1, 50:1, 20:1, 10:1, 5:1, and 1:1, *v*/*v*, each 500 mL) to produce six fractions (A–F). LC/MS analysis of the six fractions (A–F) indicated the presence of a unique molecular ion peak exhibiting a unique UV spectrum in the polar fraction E. In order to isolate the compound responsible for the unique molecular ion peak, fraction E (233 mg) was separated by preparative reversed-phase HPLC (Phenomenex Luna C_18_, 250 × 21.2 mm i.d., 5 µm, Torrance, CA, USA) using CH_3_CN/H_2_O (1:9 to 10:0, *v*/*v*, gradient system, flow rate: 5 mL/min) to yield five sub-fractions (E1–E5), which were analyzed by LC/MS to identify which fraction contained the target peak. The LC/MS data showed that the peak was detected in subfraction E5 (9.7 mg), which was then purified by semi-preparative reversed-phase HPLC (Phenomenex Luna C_18_, 250 × 10.0 mm i.d., 5 μm) using 67% MeOH/H_2_O (isocratic system, flow rate: 2 mL/min), resulting in the isolation of fridamycin A (2.0 mg, *t*_R_ = 37.5 min).

#### 2.5.1. Fridamycin A (1) 

Amorphous powder. ${\left[\alpha \right]}_{D}^{25}$ + 15.1 (*c* 0.10, MeOH); UV (MeOH) λ_max_ (log *ε*) 200 (1.24), 230 (3.56), 259 (1.32), 290 (0.87) nm; ^1^H (CD_3_OD, 800 MHz) *δ* 7.91 (1H, d, *J* = 8.0 Hz, H-3), 7.87 (1H, d, *J* = 8.0 Hz, H-4), 7.80 (1H, d, *J* = 8.0 Hz, H-8), 7.76 (1H, d, *J* = 8.0 Hz, H-7), 4.91 (1H, dd, *J* = 11.0, 1.0 Hz, H-1′), 3.70 (1H, m, H-3′), 3.63 (1H, m, H-5′), 3.45 (1H, dd, *J* = 9.0, 6.0 Hz, H-4′), 3.10 (1H, d, *J* = 13.0 Hz, H-1″a), 3.05 (1H, d, *J* = 13.0 Hz, H-1″b), 2.47 (1H, d, *J* = 15.0 Hz, H-3″a), 2.44 (1H, d, *J* = 15.0 Hz, H-3″b), 2.30 (1H, m, H-2′a), 1.40 (1H, m, H-2′b), 1.37 (3H, d, *J* = 6.0 Hz, H-6′), 1.26 (3H, s, 2″-CH_3_); ESI-MS *m/z* 485.1 [M − H]^−^ (485.1, calculated for C_25_H_25_O_10_, [M − H]^−^).

### 2.6. Cell Culture

3T3-L1 murine preadipocytes were obtained from American Type Culture Collection (ATCC, Manassas, VA, USA) and were cultured as previously described [20]. 3T3-L1 murine preadipocytes were cultured in high glucose Dulbecco’s Modified Eagle’s Medium (DMEM; Thermo Fisher Scientific Korea Ltd., Seoul, Korea) supplemented with 10% bovine calf serum (BCS; Thermo Fisher Scientific) and antibiotic-antimycotic solution (Gibco BRL, Middlesex, UK). 

### 2.7. Cell Differentiation

The methods used for differentiating 3T3-L1 preadipocytes have been described previously [20]. When 3T3-L1 murine preadipocytes reached 100% confluence, they were cultured in DMEM containing 10% fetal bovine serum (FBS; Thermo Fisher Scientific Korea Ltd., Seoul, Korea), antibiotic-antimycotic solution, 0.5 mM isobutylmethylxanthine (IBMX; Merck KGaA, Darmstadt, Germany), 1 μM dexamethasone (Sigma-Aldrich, Saint Louis, MO, USA), and 5 μg/mL insulin (Merck KGaA, Darmstadt, Germany) for 2 days (days 0–2). Cells were then maintained in DMEM supplemented with 10% FBS, antibiotic-antimycotic solution, and 5 μg/mL insulin for an additional 2 days (days 3–4) followed by culture in DMEM containing 10% FBS and antibiotic-antimycotic solution for an additional 4 days (days 5–8). 

### 2.8. Cell Viability Assay

For the measurement of cell viability, EX-Cytox (DOGEN Bio., Seoul, Korea) was used according to the manufacturer’s instructions. Differentiated 3T3-L1 adipocytes were incubated with various concentrations of fridamycin A for 48 h and then assessed at 450 nm using a microplate reader (Victor^TM^ X4, PerkinElmer, Waltham, MA, USA).

### 2.9. Glucose Uptake Assay

The methods used for measuring glucose uptake in 3T3-L1 preadipocytes have been described previously [21]. Differentiated cells were cultured in low-glucose DMEM (Gibco BRL) for 4 h and then incubated with fridamycin A or rosiglitazone (an antidiabetic drug used as a positive control [22]) in glucose-depleted DMEM (Gibco BRL) for 1 h. The cells were then treated with the fluorescent glucose indicator, 10 μM 2-[N-(7-nitrobenz-2-oxa-1,3-diazol-4-yl)amino]-2-deoxyglucose (2-NBDG; Thermo Fisher Scientific) for 30 min. After washing the cells with phosphate buffered saline (PBS), cell pellets were resuspended in PBS and passed through a 40 μm cell strainer. The fluorescence intensity of 2-NBDG at excitation/emission = 485/535 nm was measured using a fluorescence microplate reader (Victor^TM^ X4).

### 2.10. Western Blotting

Proteins were extracted using a buffer containing 25 mM HEPES, 150 mM NaCl, 1% Triton X-100, 10% glycerol, 5 mM EDTA, 10 mM NaF, 2 mM Na_3_VO_4_, and protease inhibitor cocktail (Roche Korea, Seoul, Korea). Proteins were separated by 10% sodium dodecyl sulfate-polyacrylamide gel electrophoresis and transferred to polyvinylidene fluoride membranes (Merck KGaA, Darmstadt, Germany) using a wet transfer system. Membranes were incubated overnight at 4 °C with the following primary antibodies: anti-total AMPK, anti-phosphorylated AMPK, and anti-beta-actin (AbFrontier, Seoul, Korea). Membranes were then probed with the anti-rabbit-IgG-horseradish peroxidase-conjugated secondary antibody (Santa Cruz Biotechnology, Dallas, TX, USA). Antibody–antigen complexes were detected using enhanced chemiluminescence (ECL) reagents (GE Healthcare Korea, Songdo, Korea). For the quantification of phospho-AMPK and total-AMPK, ATTO image analysis software (CS analyzer 4, Tokyo, Japan) was used.

### 2.11. Oil Red O Staining

Lipid accumulation in 3T3-L1 cells was evaluated by Oil Red O staining as described previously [20]. At day 6 after differentiation, cells were washed twice with PBS, fixed with 4% paraformaldehyde for 15 min, washed twice with PBS, then stained with filtered 0.3% Oil Red O solution in isopropanol for 30 min. After washing cells with PBS, images showing lipid accumulation were acquired using an EVOS FL Imaging System (Thermo Fisher Scientific Korea Ltd., Seoul, Korea). For the quantification of adipocyte differentiation, Oil Red O dye was eluted by incubation with isopropanol and the absorbance was measured at 490 nm using a microplate reader (Victor^TM^ X4).

### 2.12. Statistical Analysis

Statistical significance (*p* < 0.05) was determined using two-tailed unpaired *t*-tests (GraphPad Software, San Diego, CA, USA). 

## 3. Results and Discussion

### 3.1. Isolation and Identification of Fridamycin A 

LC/MS/UV-based analysis of the EtOAc fraction revealed a unique molecular ion peak at *m/z* 485.1 [M − H]^−^ with a unique UV spectrum (λ_max_ 200, 230, 259, and 290 nm) (Figure 1). LC/MS/UV-guided fractionation and repeated semi-preparative HPLC resulted in the isolation of the compound corresponding to the peak with the unique MS/UV pattern. Comparison with our in-house UV library (Figure 1B) and the molecular ion detected by LC/MS (Figure 1C and Appendix A) led us to determine the metabolite as fridamycin A (Figure 1C). To further confirm the structure, its ^1^H NMR spectrum (Appendix A) was obtained, where the proton signals for two methyls at *δ*_H_ 1.26 (3H, s) and 1.37 (3H, d, *J* = 6.0 Hz), three methylenes at *δ*_H_ 1.40 (1H, m)/2.30 (1H, m), 2.44 (1H, d, *J* = 15.0 Hz)/2.47 (1H, d, *J* = 15.0 Hz), and 3.05 (1H, d, *J* = 13.0 Hz)/3.10 (1H, d, *J* = 13.0 Hz), and four methines at *δ*_H_ 3.45 (1H, dd, *J* = 9.0, 6.0 Hz), 3.63 (1H, m), 3.70 (1H, m), and 4.91 (1H, dd, *J* = 11.0, 1.0 Hz), as well as four aromatic protons at *δ*_H_ 7.76 (1H, d, *J* = 8.0 Hz), 7.80 (1H, d, *J* = 8.0 Hz), 7.87 (1H, d, *J* = 8.0 Hz), and 7.91 (1H, d, *J* = 8.0 Hz) were clearly observed, suggesting that the compound was fridamycin A, an anthraquinone derivative linked with a sugar moiety [23,24,25]. 

### 3.2. Fridamycin A Increased Glucose Uptake into Differentiated 3T3-L1 Cells 

As insulin-sensitive tissues such as skeletal muscle and adipose tissue contribute to the maintenance of glucose homeostasis [26,27], we investigated the antidiabetic effects of fridamycin A in 3T3-L1 adipocytes. To determine suitable concentrations of fridamycin A for use in cell-based studies, differentiated 3T3-L1 adipocytes were incubated with various concentrations of fridamycin A for 48 h and cell viability was assessed using an EZ-Cytox assay kit. We found that treatment with 1 or 10 μM fridamycin A had no effect on cell viability, indicating that these concentrations were appropriate for the treatment of cells (Figure 2A). We then investigated the effect of fridamycin A on glucose uptake into differentiated 3T3-L1 adipocytes using the fluorescent glucose probe, 2-NBDG. Cells were incubated with 10 μM fridamycin A or 2 μM rosiglitazone (an antidiabetic drug used as a positive control) for 1 h. After changing the culture medium, the cells were treated with 2-NBDG for 30 min and the fluorescence intensity was measured. Fridamycin A treatment significantly increased fluorescence intensity compared with the control group, demonstrating that incubation with fridamycin A enhanced cellular glucose uptake (Figure 2B). Rosiglitazone also markedly increased cellular fluorescence, indicating that the fluorescent probe was functional in our cell systems (Figure 2B).

### 3.3. Fridamycin A Enhanced AMPK Phosphorylation in Differentiated 3T3-L1 Cells

AMP-activated protein kinase (AMPK) is an energy sensor that regulates cellular metabolism, and its activation leads to improved insulin sensitivity [28,29,30]. Previous reports showed that metformin, an effective hypoglycemic drug, increased AMPK activity in vivo, and that resveratrol, a phytoalexin found in *Polygonum cuspidatum*, mulberry, grape, and red wine, enhanced glucose uptake in C2C12 muscle cells through the activation of the AMPK pathway [7,31]. We investigated whether fridamycin A might increase AMPK phosphorylation in differentiated 3T3-L1 adipocytes. 3T3-L1 preadipocytes were differentiated in the presence of 1 or 10 μM fridamycin A for 3 days and Western blotting was performed. We found that 10 μM fridamycin A treatment induced AMPK phosphorylation compared to controls (Figure 2C). Moreover, the incubation of mature adipocytes with fridamycin A for 3 days significantly induced AMPK phosphorylation levels (Figure 2D,E). These results indicated that fridamycin A induced glucose uptake in 3T3-L1 cells by activating the AMPK signaling pathway.

### 3.4. Fridamycin A Did Not Affect Adipocyte Differentiation

We next examined the effect of fridamycin A on lipid accumulation. 3T3-L1 preadipocytes were differentiated in the presence of fridamycin A or rosiglitazone. On day 6 of differentiation, the extent of adipocyte differentiation was monitored by Oil Red O staining. Rosiglitazone has been shown to induce adipogenesis and increase weight gain in rodents and humans [32]. Consistent with these results, we found that rosiglitazone induced adipocyte differentiation (Figure 3A,B). Cryptotanshinone isolated from the dried roots of *Salvia militiorrhiza* has been shown to significantly reduce lipid droplets and exert anti-obesity effects in diet-induced obese mice [8]. In our study, treatment with fridamycin A did not affect adipocyte differentiation (Figure 3A,B), suggesting that fridamycin A stimulated glucose uptake through the activation of the AMPK signaling pathway without significant adipogenesis. This suggests that its mechanism of action is different from that of rosiglitazone. It has been reported that rosiglitazone as an anti-diabetic drug stimulates adipocyte differentiation via the activation of the peroxisome proliferator-activated receptor γ (PPAR γ) [33,34]. On the contrary, fridamycin A may play a role as an insulin sensitizer through a PPAR γ-independent signaling pathway. However, the direct target of fridamycin A that results in antidiabetic effects remains to be identified. These results suggested that fridamycin A could be a potential therapeutic candidate for the management of type 2 diabetes.

## 4. Conclusions

This study provided scientific evidence for the potential of fridamycin A for the management and treatment of diabetes. For the first time, we demonstrated that fridamycin A stimulated glucose uptake into differentiated 3T3-L1 adipocytes through the activation of the AMPK signaling pathway. The natural microbial product, fridamycin A, was isolated from *Actinomadura* sp. RB99 acquired from the surface of a termite worker (*M. natalensis*), which has been recently recognized as an untapped natural source and identified using an LC/MS/UV-based dereplication approach. In addition, fridamycin A did not affect lipid accumulation, suggesting that fridamycin A may have fewer side effects, such as weight gain, compared to rosiglitazone, a commonly used antidiabetic drug. Our results suggested that fridamycin A could be a potential therapeutic agent for the management of type 2 diabetes.

## Figures and Tables

**Figure 1 nutrients-11-00765-f001:**
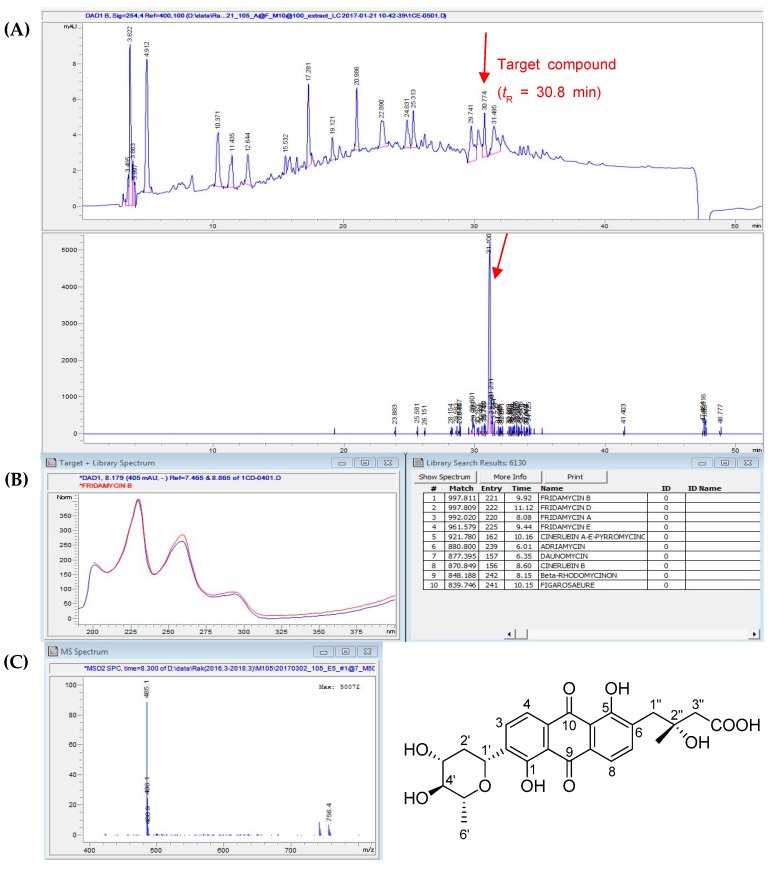
(**A**) LC/MS analysis of the EtOAc fraction (detection wavelength was set at 254 nm) and extracted ion chromatogram (EIC) for *m/z* 485.1 in negative ESI-MS mode. (**B**) Comparison of UV data of the peak at retention time 30.8 min with our in-house UV library. (**C**) Negative ion-mode ESI-MS data of the peak and the chemical structure of fridamycin A.

**Figure 2 nutrients-11-00765-f002:**
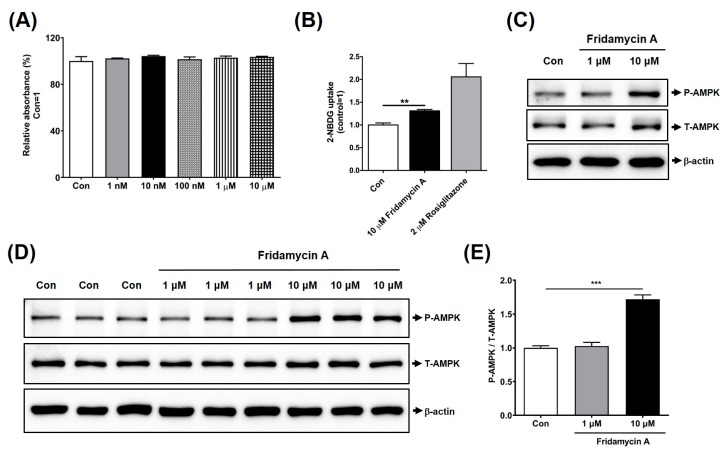
Fridamycin A increased glucose uptake in 3T3-L1 adipocytes. (**A**) 3T3-L1 adipocytes were incubated with the indicated concentrations of fridamycin A for 48 h and cell viability was determined using an EZ-Cytox assay kit. (**B**) Mature 3T3-L1 adipocytes were treated with control (Con; 0.05% dimethyl sulfoxide), 10 μM fridamycin A or 2 μM rosiglitazone for 1 h and then incubated with the fluorescent glucose indicator, 2-NBDG, for 30 min. Fluorescence intensity was measured using a fluorescence microplate reader. 3T3-L1 preadipocytes (**C**) or mature adipocytes (**D**) were incubated with fridamycin A or rosiglitazone for 3 days and analyzed by Western blot. (**E**) Quantification of phospho-AMP-activated protein kinase (AMPK) and total-AMPK was performed using ATTO image analysis software. Results are expressed as the mean ± the standard error of the mean (SEM). Data were analyzed using two-tailed unpaired *t*-tests. *** *p* < 0.001, ** *p* < 0.01 compared to the control group.

**Figure 3 nutrients-11-00765-f003:**
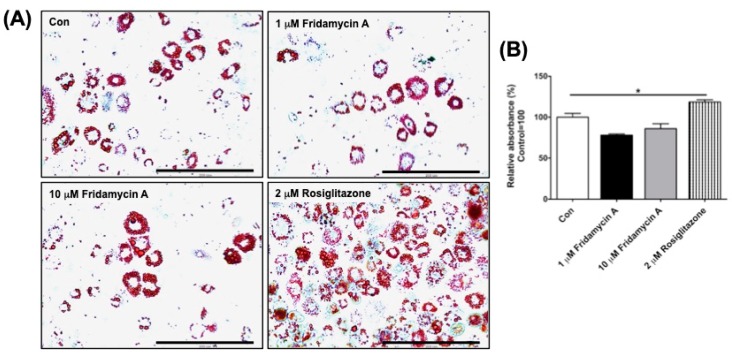
Fridamycin A did not affect adipocyte differentiation. (**A**) 3T3-L1 preadipocytes were differentiated in the presence of fridamycin A or rosiglitazone and the extent of differentiation was assessed by Oil Red O staining on day 6 of differentiation. (**B**) For the quantification of lipid accumulation, the absorbance of Oil Red O dye was measured using a microplate reader. Results are expressed as the mean ± SEM. Data were analyzed using two-tailed unpaired *t*-tests. * *p* < 0.05 compared to the control group. Scale bar: 200 μm.

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
