# Peer review of "Fridamycin A, a Microbial Natural Product, Stimulates Glucose Uptake without Inducing Adipogenesis"

_nutrients, 2019, doi:10.3390/nu11040765_

Round 1

Reviewer 1 Report

The manuscript discussed the isolation, purification, and characterization of a natural microbial product Fridamycin A from Actinomadura sp. RB99. The compound looks promising as an anti-diabetic drug although further animal studies are required. The study is straightforward and comprehensive. However, the authors should consider the following comments for minor improvement prior to re-submission. 

Line no. 123 : Mention the species for cell line used (Mouse or Human)?

Figure 1. Re-format the figure to publication quality images.

Figure 2B. There is no consistency in units? Is it uM or mM? Does not match with the text.

Figure 4. Images are too bright. Oil-Red staining is not clear. Scale bar is not clear. 

It would be conclusive if authors could show the effect of Fridamycin A on Human adipocytes cell line.

Author Response

The manuscript discussed the isolation, purification, and characterization of a natural microbial product Fridamycin A from Actinomadura sp. RB99. The compound looks promising as an anti-diabetic drug although further animal studies are required. The study is straightforward and comprehensive. However, the authors should consider the following comments for minor improvement prior to re-submission. 

Line no. 123 : Mention the species for cell line used (Mouse or Human)?

Response: We thank the reviewer for helpful comment. 3T3-L1 preadipocyte cell line is derived from mouse. According to the reviewer’s comment, we added this description to the Materials and Methods section of the revised manuscript.

Figure 1. Re-format the figure to publication quality images.

Response: We thank the reviewer for helpful guidance. We have re-formatted the Figure 1 to publication quality image, according to your suggestion and changed TIC chromatogram of the EtOAc fraction in negative ESI-MS mode to EIC chromatogram for m/z 485.1 in negative ESI-MS mode. Hopefully the revised figure would be suitable for publication.

Figure 2B. There is no consistency in units? Is it uM or mM? Does not match with the text.

Response: We thank the reviewer for helpful guidance. We made a mistake in writing the units. The unit used is μM in the study. We have corrected the mistakes in the Results and Discussion section of the revised manuscript.

Figure 4. Images are too bright. Oil-Red staining is not clear. Scale bar is not clear. 

Response: Thank you very much for the helpful comments. According to the reviewer’s comment, we adjusted the brightness of the staining to acquire more clear images. We also changed the color of scale bar to black. We replaced the staining data with the clear images in Figure 3 of the revised manuscript.

It would be conclusive if authors could show the effect of Fridamycin A on Human adipocytes cell line.

Response: We appreciate the helpful comments. 3T3-L1 murine preadipocyte cell line that we used in this study is also a well-established preadipose cell line. For this reason, we chose this cell line and performed an experiment. We also agree on your valuable opinion about the use of human adipocytes cell line. If we can obtain human adipocytes cell line, we will try to examine the effect of fridamycin A in this human cell line, however, we don’t have the human adipocytes cell line, unfortunately. Please consider our real situation. 

Reviewer 2 Report

The authors report on the biological properties of fridamycin A as antidiabetic compound.

The manuscript is interesting and well within the aims and scopes of the journal.

Yet, some major modifications are necessary before it can be accepted for publication in this journal.

These modifications are listed below one by one:

TITLE:

- Please consider to change the tense of the verb included in the title i.e. insert “stimulate”. I think this is more appropriate.

INTRODUCTION:

- Lines 53 on: Please write the complete botanical names of the fungus and plant species i.e. Beauveria bassiana (Bals.-Criv.) Vuill., Streptomyces Waksman & Henrici etc.

MATERIALS AND METHODS:

- Please provide more details about the NMR, HPLC, ESI instruments.

- Please specify the exact concentration ratios of the solvents you used for the CC procedure.

- Lines 106 on: “LC/MS analysis of the six fractions...” Is it possible that you found something only in one fraction? Were you looking for something in particular? If so, as it is clear in the following lines, please specify it also here.

- Why did you not perform also a 13C-NMR experiment? You based all your work on this compound and having a full spectral identification is extremely important and compulsory.

- IMPORTANT: The assignments of the proton and carbon compounds with the structure is fundamental and compulsory. Please do it. In this sense number the carbons in the structure.

RESULTS AND DISCUSSION:

- Paragraph 3.1.: All this part is useless since you described it in the relative paragraph of the Materials and Methods section. Instead, I would write something about the spectral identification.

- Supplementary figures are not associated with the manuscript in the system. Please check the correct uploading of the files.

- Lines 198 and 202: In the former line you wrote you studied the effect of the compound n cell variability at the μM concentration but then you performed your biological assays at mM concentrations? Is this not contradictory also given your discussion of the first results at line 199? I hope this is only a writing mistake.

Moreover, why did you not directly compare the biological activities of your compound and the control at the same concentrations?

- Line 243: “This suggests that its mechanism of action is different from that of rosiglitazone.” Could you please make an hypothesis and compare these two mechanisms?

GRAMMAR, WRITING, LANGUAGE, SPELLING:

- I noticed a couple of minor mistakes in this sense. Please check all the manuscript for these.

Author Response

The authors report on the biological properties of fridamycin A as antidiabetic compound.

The manuscript is interesting and well within the aims and scopes of the journal.

Yet, some major modifications are necessary before it can be accepted for publication in this journal.

These modifications are listed below one by one:

TITLE:

- Please consider to change the tense of the verb included in the title i.e. insert “stimulate”. I think this is more appropriate.

Response: We thank the reviewer for helpful comment. According to the reviewer’s comment, we changed the title to “Fridamycin A, a Microbial Natural Product, Stimulates Glucose Uptake without Inducing Adipogenesis”

INTRODUCTION:

- Lines 53 on: Please write the complete botanical names of the fungus and plant species i.e. Beauveria bassiana (Bals.-Criv.) Vuill., Streptomyces Waksman & Henrici etc.

Response: We thank the reviewer for helpful guidance. However, the examples that you suggested are certainly not correct way. We don’t think the scientific names of the fungus and bacteria are problematic.      

MATERIALS AND METHODS:

- Please provide more details about the NMR, HPLC, ESI instruments.

Response: We thank the reviewer for helpful comment. We have added the information for NMR instrument; 800 NMR spectrometer (Varian, Palo Alto, CA, USA) in the revised manuscript (See line 72). The information for the HPLC and ESI (LC/MS) instruments was already provided in the original manuscript; HPLC (Waters Corporation, Milford, CT, USA) and (Shimadzu, Tokyo, Japan) as well as LC/MS (Agilent Technologies, Santa Clara, CA, USA).

- Please specify the exact concentration ratios of the solvents you used for the CC procedure.

Response: We thank the reviewer for helpful comment. We have added the exact information for solvent that we have used for column chromatography (See lines 108 and 113).

- Lines 106 on: “LC/MS analysis of the six fractions...” Is it possible that you found something only in one fraction? Were you looking for something in particular? If so, as it is clear in the following lines, please specify it also here.

Response: We thank the reviewer for helpful comment. We have added the description of our purpose to isolate the target compound in the following lines (See lines 110-111).

- Why did you not perform also a 13C-NMR experiment? You based all your work on this compound and having a full spectral identification is extremely important and compulsory.

Response: We thank the reviewer for helpful guidance. We agree on your valuable opinion about the full spectral identification for fridamycin A. However, fridamycin A is a known compound and its full NMR data were already reported in the literatures [23-25]. Thus, we did not perform the 13C NMR experiment. Actually, the key point of this study was that fridamycin A was identified from the MeOH extracts of Actinomadura sp. RB99 by a comparative LC/MS-based analytical approach, and its structure was easily confirmed by comparison with our in-house UV library and the molecular ion detected in LC/MS.

<References>

[23] Kusumi, S.; Tomono, S.; Okuzawa, S.; Kaneko, E.; Ueda, T.; Sasaki, K.; Takahashi, D.; Toshima, K. Total Synthesis of vineomycin B2. J. Am. Chem. Soc. 2013, 135, 15909-15912.

[24] Sezaki, M.; Kondo, S.; Maeda, K.; Umezawa, H. The structure of aquayamycin. Tetrahedron 1970, 26, 5171-5190.

[25] Alvi, K.A.; Baker, D.D.; Stienecker, V.; Hosken, M.; Nair, B.G. Identification of inhibitors of inducible nitric oxide synthetase from microbial extracts. J. Antibiot. 2000, 53, 496-501.

- IMPORTANT: The assignments of the proton and carbon compounds with the structure is fundamental and compulsory. Please do it. In this sense number the carbons in the structure.

Response: We thank the reviewer for helpful guidance. We have added the NMR assignment for proton (See 2.5.1. section) and carbon numbering in the structure of Figure 1. In addition, we have uploaded the revised 1H NMR data of fridamycin A by adding NMR assignment in Supplementary Material.  

RESULTS AND DISCUSSION:

- Paragraph 3.1.: All this part is useless since you described it in the relative paragraph of the Materials and Methods section. Instead, I would write something about the spectral identification.

Response: We thank the reviewer for helpful guidance. According to your suggestion, we have modified the part of 3.1. section by removing the redundant repetitive content and instead, adding NMR spectral identification for fridamycin A, together with the related references.

<References>

[24] Sezaki, M.; Kondo, S.; Maeda, K.; Umezawa, H. The structure of aquayamycin. Tetrahedron 1970, 26, 5171-5190.

[25] Alvi, K.A.; Baker, D.D.; Stienecker, V.; Hosken, M.; Nair, B.G. Identification of inhibitors of inducible nitric oxide synthetase from microbial extracts. J. Antibiot. 2000, 53, 496-501.

- Supplementary figures are not associated with the manuscript in the system. Please check the correct uploading of the files.

Response: Supplementary Material contains ESIMS and 1H NMR data of fridamycin A, microbial material, DNA extraction and PCR amplification, and sequencing and species identification, which are the important information in the study. We have also corrected the 1H NMR data of fridamycin A by adding NMR assignment in Supplementary Material.

- Lines 198 and 202: In the former line you wrote you studied the effect of the compound n cell variability at the μM concentration but then you performed your biological assays at mM concentrations? Is this not contradictory also given your discussion of the first results at line 199? I hope this is only a writing mistake.

Response: We thank the reviewer for helpful comment. Actually, the points were our inadvertent mistakes. The unit used is μM in the study. We have corrected the mistakes in the Results and Discussion section of the revised manuscript.

Moreover, why did you not directly compare the biological activities of your compound and the control at the same concentrations?

Response: We thank the reviewer for helpful guidance. When we screened various concentrations of fridamycin A to select appropriate concentrations, 0.1 or 1 μM fridamycin A did not affect the biological activities tested (stimulation of glucose uptake). On the contrary, we found that treatment with 10 μM fridamycin A increased glucose uptake. For this reason, 10 μM fridamycin A was selected for further study. In case of rosiglitazone, 2 μM treatment works in cell-based studies.

- Line 243: “This suggests that its mechanism of action is different from that of rosiglitazone.” Could you please make an hypothesis and compare these two mechanisms?

Response: We thank the reviewer for helpful comment. It has been reported that rosiglitazone as an anti-diabetic drug stimulates adipocyte differentiation via the activation of peroxisome proliferator-activated receptor γ (PPAR γ) [33,34]. On the contrary, fridamycin A may play a role as insulin sensitizer through PPAR γ-independent signaling pathway. According to the suggestion, we have added this description to the Results and discussion section of the revised manuscript, together with the related references.

<References>

[33] Minge, C.E.; Bennett, B.D.; Norman, R.J.; Robker, R.L. Peroxisome proliferator-activated receptor-gamma agonist rosiglitazone reverses the adverse effects of diet-induced obesity on oocyte quality. Endocrinology 2008, 149, 2646-2656.

[34] Fryer, L.G.; Parbu-Patel, A.; Carling, D. The anti-diabetic drugs rosiglitazone and metformin stimulate AMP-activated protein kinase through distinct signaling pathways. J. Biol. Chem. 2002, 277, 25226-25232.

GRAMMAR, WRITING, LANGUAGE, SPELLING:

- I noticed a couple of minor mistakes in this sense. Please check all the manuscript for these. Response: We thank the reviewer for helpful comment. According to your suggestion, we have reviewed whole revised manuscript and corrected several mistakes.

Round 2

Reviewer 2 Report

The authors presented a revised version of the work I had previously reviewed.

The manuscript has been well improved but I still have got one important concern.

- My Query: Please provide more details about the NMR, HPLC, ESI instruments.

Your Response: We thank the reviewer for helpful comment. We have added the information for NMR instrument; 800 NMR spectrometer (Varian, Palo Alto, CA, USA) in the revised manuscript (See line 72). The information for the HPLC and ESI (LC/MS) instruments was already provided in the original manuscript; HPLC (Waters Corporation, Milford, CT, USA) and (Shimadzu, Tokyo, Japan) as well as LC/MS (Agilent Technologies, Santa Clara, CA, USA).

Reply: This part was not satisfied. It is necessary that you describe the HPLC and HPLC-MS details i.e. carrier gas, temperature, nebulizer, volume of injection, time of the analysis, voltages etc so that your analysis may be redone in the future by someone else.

Author Response

The authors presented a revised version of the work I had previously reviewed.

The manuscript has been well improved but I still have got one important concern.

- My Query: Please provide more details about the NMR, HPLC, ESI instruments.

Your Response: We thank the reviewer for helpful comment. We have added the information for NMR instrument; 800 NMR spectrometer (Varian, Palo Alto, CA, USA) in the revised manuscript (See line 72). The information for the HPLC and ESI (LC/MS) instruments was already provided in the original manuscript; HPLC (Waters Corporation, Milford, CT, USA) and (Shimadzu, Tokyo, Japan) as well as LC/MS (Agilent Technologies, Santa Clara, CA, USA).

Reply: This part was not satisfied. It is necessary that you describe the HPLC and HPLC-MS details i.e. carrier gas, temperature, nebulizer, volume of injection, time of the analysis, voltages etc so that your analysis may be redone in the future by someone else.

Response: We thank the reviewer for helpful comment. Actually, we have misunderstood what you want to revise. Now, we have added the detailed information about what you want in the revised manuscript. Hopefully the revision would be satisfactory.